# The effect of voluntary disclosure on financial performance: Empirical study on manufacturing industry in Indonesia

Meiryani[1]*, Shi Ming Huang[2], Gatot Soepriyanto[1], Jessica[1], Mochammad Fahlevi[3], Sandra Grabowska[4], Mohammed Aljuaid[5]

1 Accounting Department, School of Accounting, Bina Nusantara University, Jakarta, Indonesia,
2 Accounting and Information Technology Department, National Chung Cheng University, Chiayi County, Taiwan, 3 Management Department, BINUS Online Learning, Bina Nusantara University, Jakarta, Indonesia, 4 Silesian University of Technology, Silesia, Poland, 5 Department of Health Administration, College of Business Administration, King Saud University, Riyadh, Saudi Arabia

* meiryani@binus.edu

**Data Availability Statement:** Data are available from the www.idx.co.id. Institutional Data Access / Ethics Committee in Tower 1. Jl. Jend. Sudirman Kav 52-53 Jakarta Selatan 12190, Indonesia. Contact Number Toll Free: 0800-100-9000.

## Abstract

The manufacturing sector's adherence to managing natural resources from the environment still needs to be improved. This study's objective is to determine how Corporate Social Responsibility (CSR) influences the financial performance of manufacturing firms featured in the LQ45 Index, as measured by Return on Assets (ROA), Return on Equity (ROE), and Net Profit Margin (NPM). All manufacturing companies that are included in the LQ45 Index's population for this study were sampled using the purposive sampling method. This study uses secondary data from the CSRI based on the Global Reporting Initiative (GRI) G4 standard for 2018–2020 and the annual reports of companies in the manufacturing industry sector listed on the LQ45 Index. Moreover, applying a quantitative methodology, descriptive statistical methods, conventional assumption tests, and simple linear regression analysis were applied in this study's data analysis. The results of the study proved that CSR has a significant effect on ROA but does not affect the ROE and NPM of LQ45 manufacturing companies. In accordance with the signaling theory, CSR disclosure sends a favourable message to outsiders, which stakeholders and shareholders will respond to through changes in business earnings. CSR implementation can establish a positive image for the company, but it can also improve the company's image in both the commodity and capital markets. Investors will be more attracted to a company with a positive corporate image since a positive corporate image increases consumer loyalty. As consumer loyalty rises, the company's sales will likewise rise, and its profitability will increase as a result. This paper opens a new research path in corporate social responsibility and financial performance for possible links among variables; a matter that has not been previously explored in Indonesia Manufacturing Public Companies.

## 1. Introduction

Technological developments marked by the emergence of Industry 4.0 have an impact on the company's operational activities so that companies not only have the goal of seeking profit but also must prioritize the needs of the community because of social inequality and preserve the

**Funding:** The authors would like to extend their appreciation to King Saud University for funding this work through the researcher supporting project (RSP2023R481), King Saud University, Riyadh, Saudi Arabia.

**Competing interests:** The authors have declared that no competing interests exist.

environment due to environmental damage [1]. This is due to the company's close relationship with residents and the surrounding environment, both directly and indirectly. To show the results of company management that produce positive benefits for the company itself, society, and the environment, it is necessary to have corporate social responsibility (CSR) [2]. CSR itself can have a positive effect because the company can improve its reputation by increasing public trust and concern for the environment so that there is an increase in product sales that generate profits for the company. Increased company profits can attract investors' interest in their investment decisions that affect the company's financial performance [3]. Although many people say that CSR is good, socialization efforts are still mandatory so that companies better understand the concept of CSR and its functions. Although many companies contribute to the country's economic development, many companies also exploit natural resources, which results in severe environmental damage [4].

In recent years, there has been a phenomenon of increasing environmental damage, such as waste disposal, air pollution, deforestation, development systems that are not friendly to the environment, and climate change [5]. This phenomenon warns residents that managing limited natural resources is crucial, so companies must be able to manage these resources efficiently, especially in carrying out their operational activities. According to the Ministry of Environment and Forestry [6], the manufacturing sector's compliance with managing natural resources from the environment is still low. Until 2019, the number of companies that registered themselves to be assessed for compliance was quite low, only reaching 597 companies or 29.15 percent. According to the Ministry of Finance [7], manufacturing activity weakened during the application of the Indonesia's large-scale social restrictions rules because of the COVID-19 pandemic. The decline in manufacturing activity occurred in sales or production [8]. The decrease in sales activity increased excess capacity, which was shown to hamper the recruitment of workers. The company also decreased its purchasing and stocking activities for reasons of efficiency and the pressure on costs caused by the depreciation of the exchange rate and lower selling prices. Some companies also provide discounts so that the goods are sold.

Kravchenko et al. [9] discovered that manufacturing companies are closely related to social and environmental concerns, or in the sense that they have the most stakeholders. The same thing was stated by Prieto-Sandoval et al. [10], who provided support with their statement, which stated that manufacturing companies disclose information significantly more than other types of industries. Manufacturing companies are believed to need a better image from society because they are vulnerable to political influence and criticism from social activists, so it is assumed that manufacturing companies will provide broader corporate social responsibility disclosures than non-manufacturing companies [11]. A manufacturing company is a company that processes raw materials to turn them into goods ready for market, which involves various sources of raw materials, production processes, and technology [12–14].

By examining the ratios, one of which is the profitability ratio, which is a crucial factor for investors, the outcomes of the company's financial performance may be seen in the financial statements [15]. The ability of the company to pursue profit over a specific period is measured by the profitability ratio. This ratio gives an indication of a company's management's efficiency based on its sales profit or investment income [16]. When a company's profitability rises, both the public's and investors' perceptions of the company's value in terms of its efficacy will rise [17]. Due to the importance of CSR being carried out by companies, this study was carried out to describe whether there is an influence of CSR on financial performance in manufacturing companies measured by Return on Asset (ROA), Return on Equity (ROE), and Net Profit Margin (NPM) where CSR measurements are based on Global Reporting Initiative (GRI) G4 standards.

This research focuses on manufacturing companies. The reason for choosing a manufacturing company is because manufacturing companies cannot be separated from society as their external environment, because manufacturing companies have more influence or impact on the surrounding environment as a result of the activities carried out by the company and fulfill all aspects of the theme of corporate social responsibility disclosure.

Based on the background, the research problems are:

- Does Corporate Social Responsibility have a significant effect on Return on Assets?

- Does Corporate Social Responsibility have a significant effect on Return on Equity?

- Does Corporate Social Responsibility have a significant effect on Net Profit Margin?

This study intends to determine how CSR influences the financial performance of manufacturing businesses listed in the LQ45 Index for the 2018–2020 timeframe, as indicated through ROA, ROE, and NPM data analysis. The research used in this study has certain limitations because it does not extend upon and is more narrowly focused on the research objectives that have been described:

- Only discusses the effect of CSR on financial performance, which means this research will not discuss other factors.

- Conducted on manufacturing industry companies, which means this research will not discuss other industrial companies.

The remainder of this paper is organized as follows: Section 2 presents the theoretical background and develops the hypotheses. Section 3 describes the methods and data sources. Section 4 reports the empirical results; Section 5 presents a related discussion; the last section concludes the paper and offers suggestions and directions for future research.

## 2. Literature review

### Corporate social responsibility

According to the World Business Council and Sustainability Development [18], CSR is defined as a form of the company's ongoing commitment to contribute to economic development and behave ethically in terms of improving the quality of life of the workforce and the people in a way that is useful for the company itself and development. Understanding CSR can mean a company's commitment to support sustainable economic growth by focusing on CSR and maintaining a balance between economic, environmental, and social factors [19]. Corporate Social Responsibility Disclosure (CSRD) is a report prepared by a company to be able to communicate to interested parties regarding non-financial company activities, such as social activities, so that with this reporting it is hoped that the company can grow sustainably. CSR is the Company's responsibility and concern for the environment and society, which is carried out in a sustainable, measurable, and transparent manner, in the economic, health, education, and environmental fields, for the common interest of the company and stakeholders. Therefore, good CSR implementation will show a form of corporate concern, carried out through sustainable programs and for the common good of the company and society [20].

### Legitimacy theory

In legitimacy theory, every company wants to receive good legitimacy from the people. Companies can obtain this by carrying out CSR activities for local residents, because the goal of CSR is to build a good reputation in the eyes of the people so that the company's survival can

be guaranteed [21]. CSR helps companies to have positive legitimacy and avoid negative legitimacy at the same time because, by doing CSR, companies also automatically do positive things and do not do negative things [22].

### Agency theory

Agency theory is used to explain the relationship between the agent (the party receiving the authority) and the principal (the party giving the authority) [23], which is constructed so that the company's goals can be realized as effectively as possible [24]. According to this agency hypothesis, everyone acts in their own best interests. Due to the possibility that the agent would not always behave in the principal's best interests, there could be a conflict of interest between the principal and the agent, leading to agency fees. The disclosure of CSR in agency relations is influenced by three factors: monitoring costs, contracting costs, and political visibility [25].

### Political theory

The political theory is known as a corporate citizenship group [26]. This group of theories focuses on the interactions and relationships between corporations and society, as well as the roles and positions of corporations and their responsibilities. The company is part of society, and its position creates a social impact. As part of a social institution, companies must use their position responsibly. Companies are not only legally and morally responsible for their activities but also socially responsible, namely being good citizens.

### Instrumental theory

This theory is better known as the shareholder theory [27]. CSR appears only as a tool to achieve the economic goals of a company, which are to make a profit. The only CSR is to increase profits as much as possible for shareholders. Only the economic aspects of the interactions between companies and communities are considered [28].

### Stakeholder theory

It is a theory that the survival of a company cannot be separated from the roles of stakeholders [29], either internally or externally. CSR can be defined as a company's effort to meet these interests by addressing non-financial issues related to the company's social and environmental impacts resulting from its operational activities. Better CSR disclosure by the company will result in full support from stakeholders for the company regarding all its activities aimed at increasing performance and achieving profit [30]. The theory also states that companies are not only established for operational activities for their own sake but also need to convey benefits to other stakeholders [31].

### Signaling theory

Indicating that the company provides a signal or frequency to investors with the intent of increasing the firm's value [32]. In addition to required financial disclosures, the company also makes voluntary disclosures. CSRD made by the company in its annual report constitutes voluntary disclosure. This CSRD signifies that the company is sending a positive signal to parties outside the company, to which stakeholders and shareholders will respond with changes in stock prices and company profits [33]. This theory indicates that a company with high quality will send a signal to its stakeholders so that they can learn the company's good and bad news.

This theory is used to minimize differences so that all stakeholders receive identical or comparable information [34].

## Financial performance

Financial performance is the final result of accounting activities or the company's accounting cycle, reflecting the company's financial condition and the results of its operations [35]. The results are presented in financial statements [15]. ROA measures a company's ability to generate a net profit from its total assets. ROE measures a company's ability to generate a net profit with its capital. NPM refers to a company's ability to generate a profit margin on sales by comparing profit after tax to net sales [36]. Financial performance is an important aspect of CSR for manufacturing companies in Indonesia because it can have a significant impact on a company's long-term sustainability and its ability to continue to operate and create positive social and environmental results [37]. CSR activities can require significant financial investment, and a company's financial performance determines its ability to allocate resources to such initiatives [38]. Financially stable companies are more likely to invest in CSR initiatives that can have a positive impact on society and the environment [39]. A company's financial performance is a key indicator of its overall health, and investors, customers, and other stakeholders often use financial metrics to evaluate a company's performance. Positive financial performance can enhance a company's reputation and help build trust with stakeholders, which in turn can benefit a company's CSR initiatives [40].

There are CSR initiatives that involve regulatory compliance, and companies struggling with financial performance may face challenges meeting these requirements. Financial stability can help a company ensure that it complies with relevant regulations and avoids potential legal and reputational risks [41]. CSR initiatives often involve long-term planning and investment, and a company's financial performance can affect its ability to sustain these initiatives over time [42]. Companies with strong financial performance are better positioned to invest in sustainable CSR initiatives that create long-term value for society and the environment. Financial performance is an important aspect of CSR for manufacturing companies in Indonesia because it can influence their ability to invest in and sustain initiatives that create positive social and environmental results, enhance their reputation and build trust with stakeholders, comply with relevant regulations, and ensure long-term sustainability [11].

## Corporate Social Responsibility Index (CSRI)

CSRD is measured and expressed in the Corporate Social Responsibility Index (CSRI). The CSR measurement is carried out by assessing each item disclosed in the company's year-end report or sustainability report in accordance with the GRI guidelines, which consist of 3 main categories: 9 economic indicators (CSRI1), 34 environmental indicators (CSRI2), and 48 social indicators (CSRI3). These categories are rated 1 if disclosed and evaluated 0 if not disclosed, and then the value of each item is added up to obtain the overall CSR value of a company and compared with the GRI G4 reporting standard guidelines. The formula is:

$$CSRI_j = \frac{\sum xij}{nj}$$

: CSRI per company (j) category
: score 1 if item i is disclosed; score 0 if otherwise
: number of items for company (j), nj = 91

## Research framework

Based on the literature review which has been described previously, the variables related to this research can be formulated through a research framework in Fig 1:

The independent variable (X) in this study is CSR as measured by CSRI. The dependent variable (Y) in this study is financial performance as indicated by ROA, ROE, and NPM.

## Research hypothesis

The correlation of CSR with financial performance is explained by the three theories described previously, indicating a close relationship between the company and external parties, both of which influence each other when companies share information about social responsibility with citizens; the legitimacy and reputation of the company in the eyes of the public will increase, which will have a positive effect on the company's profitability [43]. According to research by Purnaningsih [44], CSR has a positive and statistically significant effect on ROA and ROE but no effect on Return on Sales (ROS). According to research by Erinos and Yurni-wati [45], CSR has no effect on the financial performance of manufacturing firms. According to the conclusion of Purwabangsa et al. [37], CSR has a significant positive effect on ROA, ROE, and Price to Book Value (PBV).

Companies that do not engage in CSR are more likely to face protests or demonstrations from the community, which can result in the cessation of a company's operational activities, resulting in losses. On the other hand, companies that engage in CSR can avoid protests from the public, allowing them to continue to operate effectively and achieve their goals. overall gain. Implementing CSR can reduce operational costs, increase sales volume and market share, create a positive image that attracts potential investors, and so on. It is hoped that by engaging in CSR activities, the company will be able to achieve its primary objective of seeking profit without ignoring the interests of stakeholders and environmental sustainability as a form of responsibility for the impacts its operational activities have caused [46]. The results of research by Rosiliana et al. [47] show that CSR has a negative effect or has an inverse and insignificant relationship to ROE, but a positive and significant influence on ROA, and ROS.

Companies with outstanding CSR performance can reduce capital constraints. This is due to numerous factors. First, outstanding CSR performance indicates greater stakeholder engagement, reduces the likelihood of short-term opportunistic behavior, and consequently

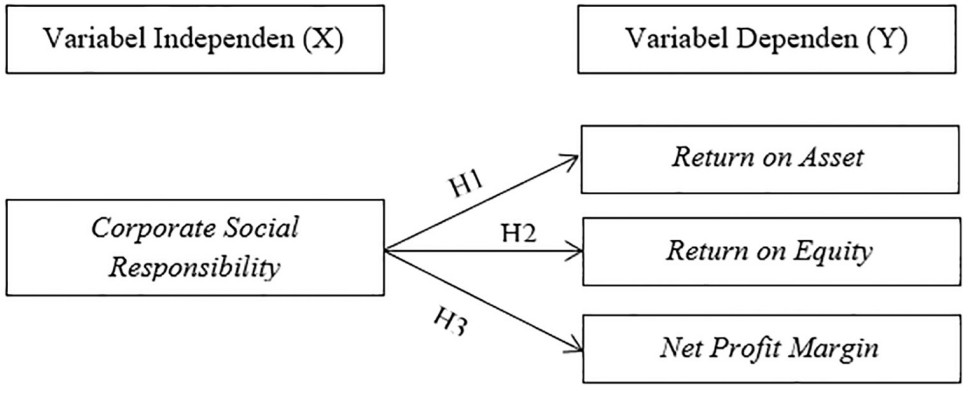

**Fig 1. Research framework.**

lowers overall contract costs. Second, companies with outstanding CSR performance are more likely to disclose their CSR activities to the market in order to signal their long-term focus and demonstrate their competitive advantages. In addition, CSR reporting can generate positive feedback, such as increased transparency of companies' social and environmental impacts and organizational structures and can alter the internal control system to improve regulatory compliance and reporting reliability [48].

In accordance with the research framework that has been presented previously, this study has several hypotheses that were tested:

- H1 = CSR has a significant effect on ROA.

- H2 = CSR has a significant effect on ROE.

- H3 = CSR has a significant effect on NPM

## 3. Research methodology

### Population and sample

The population used in this study are all manufacturing companies listed on the LQ45 Index. Based on previous findings manufacturing companies have close relationships with the community and the surrounding environment [8, 12, 13], and it is concluded that manufacturing companies have a broad scope of stakeholders. In addition, manufacturing companies carry out activities that can produce finished products by processing raw materials, so society and the environment have an important role in the survival of the company [49].

The purposive sampling method was used for research selection and sampling [50]. The reason for using this purposive sampling technique is because it is in accordance with this research, namely using quantitative research, and refers to previous research and studies that do not generalize [51]. This method is done by selecting a sample that is not random using certain criteria. The criteria used were: companies in the manufacturing industry sector listed on the LQ45 Index, disclosing consecutive annual reports, disclosing CSR in annual reports, publishing sustainability reports, and generating positive profits in a row during 2018–2020.

Based on the selection of research samples conducted by the purposive sampling method, we obtained 13 samples of manufacturing companies with the following details in Table 1:

Table 1. Manufacturing companies.

| Number | Code | Name |
| --- | --- | --- |
| 1 | ASII | Astra International Tbk |
| 2 | CPIN | Charoen Pokphand Indonesia Tbk |
| 3 | GGRM | Gudang Garam Tbk |
| 4 | HMSP | HM Sampoerna Tbk |
| 5 | ICBP | Indofood CBP Sukses Makmur Tbk |
| 6 | INDF | Indofood Sukses Makmur Tbk |
| 7 | INKP | Indah Kiat Pulp & Paper Tbk |
| 8 | INTP | Indocement Tunggal Prakarsa Tbk |
| 9 | KLBF | Kalbe Farma Tbk |
| 10 | SMGR | Semen Indonesia (Persero) Tbk |
| 11 | SRIL | PT Sri Rejeki Isman Tbk |
| 12 | TPIA | PT Chandra Asri Petrochemical Tbk |
| 13 | UNVR | Unilever Indonesia Tbk |

## Data collection method

The source of data collection in this study is secondary data (data obtained from other parties) obtained from the 2018–2020 annual reports of manufacturing industry companies listed on the LQ45 Index and CSRI based on GRI G4 standards. Data were collected using documentation study techniques or literature studies by re-recording or documenting the data listed on the website of the manufacturing company that was used as the research sample. From the documentation obtained, these data can be used to calculate ROA, ROE, and NPM. In this study, we used secondary data in the form of annual financial reports and sustainability reports. Annual financial and sustainability reports can be accessed through the website, www. idx.co.id. The sampling technique uses a purposive sampling technique to obtain a representative sample according to predetermined criteria. The data collection technique in this study was the documentation method, namely, the method of collecting data using documents related to research [52].

## Operational and measurement of research variables

**Independent variable.** CSR is the company's commitment to contribute to economic development by taking corporate social responsibility and the balance of economic, social, and environmental considerations into account. The company's CSR activities are reflected in the annual report's CSRD. This study employs content analysis based on the GRI 4 instrument with indicators of 79 items from six categories, namely economic, environmental, labor, human rights, social, and product, to measure the level of CSRD. The information used in this study regarding the Corporate Social Disclosure Index (CSDI) based on GRI was obtained from the website www.globalreporting.org. As the basis for sustainability reporting, the GRI includes three disclosure focuses: economic, environmental, and social. Gallego-Álvarez and Quina-Custodio [53] says that the three areas of focus on disclosure are broken down into six indicators: indicators of economic performance, indicators of environmental performance, indicators of labor performance, indicators of human rights, indicators of social performance, and indicators of product performance. CSDI calculations are done using dummy variables and a dichotomous approach, where each CSR item in the research instrument is given a value of 1 if it is disclosed and 0 if it is not. The total score for each company is calculated by adding the scores for each item. Each disclosure item will receive a score of 1 if it is disclosed and a score of 0 if it is not disclosed. The CSDI is measured by a ratio scale, namely the ratio of the index's value to its standard deviation:

$$\mathbf{CSDI = Xij/n}$$

Information:
CSDI = CSR Disclosure Index
Xij = Number of company disclosures, n < 79
N = number of checklist disclosure items, n = 79

**Dependent variable.** Financial performance as the dependent variable in this research model uses 3 financial ratios namely ROA, ROE, and NPM. ROA captures the effectiveness with which a business converts its assets into cash flow [36]. The ratio of net income to total assets is determined. If a company has a high return on assets, that means it is making more money with each dollar it has. A low ROA could mean that a company is not making good use of its resources. ROE captures the profit that is earned in relation to the amount of capital contributed by shareholders. Profit after tax as a percentage of stockholders' equity is computed. Investors should take heart from a high ROE, as it indicates that the business is making money for every dollar invested by shareholders. When ROE is low, it may mean that a company is

not making enough money to justify shareholder investment. NPM indicates how much of a company's revenue is kept as profit. Net income as a percentage of sales is what this function measures. If the NPM is high, that means the company is profitable relative to its revenue, which is good news for shareholders. If a company's NPM is low, it could mean that it is having trouble keeping costs in check or is competing in a very tough market.

$$\text{ROA} = \frac{net\ income}{total\ of\ the\ assets}$$

$$\text{ROE} = \frac{net\ income}{total\ of\ the\ equity}$$

$$\text{NPM} = \frac{net\ income}{net\ sales}$$

## Data analysis method

This study used a quantitative approach because the data was taken and used in the form of numbers calculated using statistical methods [52]. This was done to test the research hypothesis. Data analysis was carried out in this study using descriptive statistical analysis methods, classical assumption tests, and simple linear regression analysis [50]. A classical assumption test was conducted to determine the distribution of the data. The regression analysis was conducted to find out how the independent variable affects the dependent variable. The regression model used is based on the hypothesis to be tested:

$$\text{Y}_1 = \alpha + \beta x + \varepsilon$$

$$\text{Y}_2 = \alpha + \beta x + \varepsilon$$

$$\text{Y}_3 = \alpha + \beta x + \varepsilon$$

$\text{Y}_1, \text{Y}_2, \text{Y}_3$ = ROA, ROE, NPM
= constant
= regression coefficient
= CSR
= residue

# 4. Results

## Descriptive statistics

Descriptive statistical analysis was used to describe the data statistically. In this analysis, data descriptions of each research variable from 2018–2020 have been processed and seen from the average or mean value, standard deviation value, minimum value, and maximum value.

Based on the results of the descriptive statistical analysis in Table 2, it shows that from 39 samples (13 companies x 3 years) that have been collected:

1. CSRI variable has an average value of 0.1603269; the standard deviation value is 0.0746895; the minimum value of 0.043956; and the maximum value is 0.3186813. The minimum value of 0.043956 comes from the CSRI of the GGRM company in 2019, while the maximum value of 0.3186813 is obtained from the CSRI of the INTP company in 2018. From

**Table 2. Descriptive statistical analysis results.**

| Variable | Obs | Mean | Std. Dev. | Min | Max |
|---|---|---|---|---|---|
| CSRI | 39 | 0.1603269 | 0.0746895 | 0.043956 | 0.3186813 |
| ROA | 39 | 0.1135007 | 0.1005301 | 0.0068518 | 0.4467457 |
| ROE | 39 | 0.2379512 | 0.3386277 | 0.0134282 | 1.451084 |
| NPM | 39 | 0.1029214 | 0.0394906 | 0.0125716 | 0.2172384 |

Source: output from STATA, 2022

these data it can be concluded that the 2019 GGRM CSRI is the lowest CSRI and the 2018 INTP CSRI is the highest of the 39 samples available.

2. ROA variable has an average value of 0.1135007; the standard deviation value is 0.1005301; the minimum value of 0.0068518; and the maximum value is 0.4467457. The minimum value of 0.0068518 comes from the ROA of the TPIA company in 2019, while the maximum value of 0.4467457 is obtained from the ROA of the UNVR company in 2018. From these data it can be concluded that the ROA of TPIA 2019 is the lowest ROA and the ROA of UNVR 2018 is the highest of the 39 samples available.

3. ROE variable has an average value of 0.2379512; the standard deviation value is 0.3386277; a minimum value of 0.0134282; and a maximum value of 1.451084. The minimum value of 0.0134282 comes from the ROE of the TPIA company in 2019, while the maximum value of 1.451084 is obtained from the ROE of the UNVR company in 2020. From these data it can be concluded that the ROE of TPIA 2019 is the lowest ROE and the ROE of UNVR 2020 is the highest of the 39 samples available.

4. NPM variable has an average value of 0.1029214; the standard deviation value is 0.0394906; the minimum value of 0.0125716; and the maximum value is 0.2172384. The minimum value of 0.0125716 comes from the 2019 NPM of the TPIA company, while the maximum value of 0.2172384 is obtained from the 2018 UNVR company's NPM. From these data it can be concluded that the 2019 TPIA NPM is the lowest NPM and the 2018 UNVR NPM is the highest of the 39 samples available.

5. From all the conclusions stated above, it is known that the TPIA company has the lowest ROA, ROE, and NPM in 2019; UNVR company has the highest ROA and NPM in 2018 and has the highest ROE in 2020.

## Classical assumption test

**Normality test.** The normality test is conducted to determine whether the data were normally distributed or not [50]. In this study, the normality test was carried out using skewness and kurtosis normality test [51].

From the results of the skewness and kurtosis normality test in Table 3, that has been carried out, it is concluded that only the CSRI and NPM variables are normally distributed because the CSRI p-value 0.1983 and NPM 0.0933 are greater than the 0.05 significance level, while the ROA and ROE variables are not because they have a smaller p-value. from a significance level of 0.05, namely 0.0002 and 0.000. These results are also supported by the normal distribution graph shown in Figs 2–5. The four figures both show how the data distribution occurs with points. These data points prove that only CSRI and NPM variables have normal data distributions because the data points spread by following and surrounding the diagonal

**Table 3. Skewness and Kurtosis normality test results.**

| Variable | Obs | Pr (Skewness) | Pr (Kurtosis) | Adj chi2 (2) | Prob>chi2 |
|----------|-----|---------------|---------------|--------------|-----------|
| CSRI | 39 | 0.2190 | 0.2208 | 3.24 | 0.1983 |
| ROA | 39 | 0.0001 | 0.0091 | 16.84 | 0.0002 |
| ROE | 39 | 0.0000 | 0.0001 | 30.83 | 0.0000 |
| NPM | 39 | 0.1380 | 0.1101 | 4.74 | 0.0933 |

Source: output from STATA, 2022

line. ROA and ROE variables do not have a normal data distribution because the data points spread away from and not around the diagonal line.

Because the ROA and ROE variables are not normally distributed, data transformation must be carried out if we do not want to delete extreme data. After transforming the data with natural logarithms, the results of the skewness and kurtosis normality test are:

From Table 4, the data transformation process carried out, it can be seen that the p-value of ROA is greater than the 0.05 significance level, so it can be said that the variable has a normal data distribution. Although the p-value of ROE is still smaller than 0.05, when compared with a significance level of 0.01, the variable is normally distributed after being tested by the skewness and kurtosis normality test. In addition, according to Figs 6 and 7 after being transformed, the ROA and ROE variables have normal data distributions because the data points spread by following and surrounding the diagonal line.

**Multicollinearity test.** A multicollinearity test was conducted to determine whether there was a relationship/correlation between the independent variables or not [50]. In this study, the independent variable is only 1, which means it cannot be tested for multicollinearity.

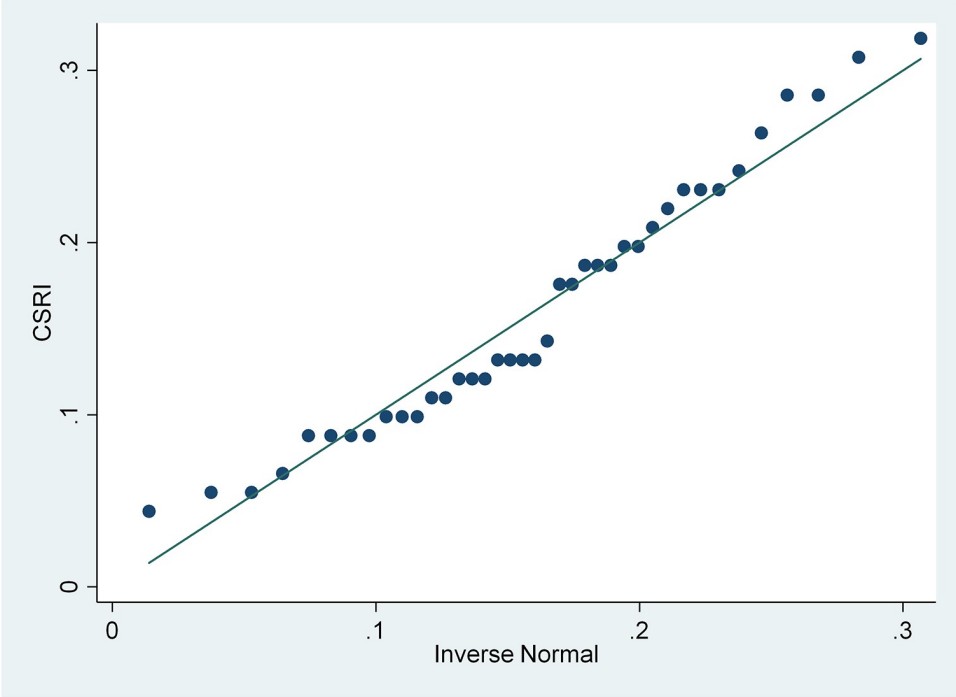

**Fig 2. Normal quantile plot (CSRI).** Source: output from STATA, 2022.

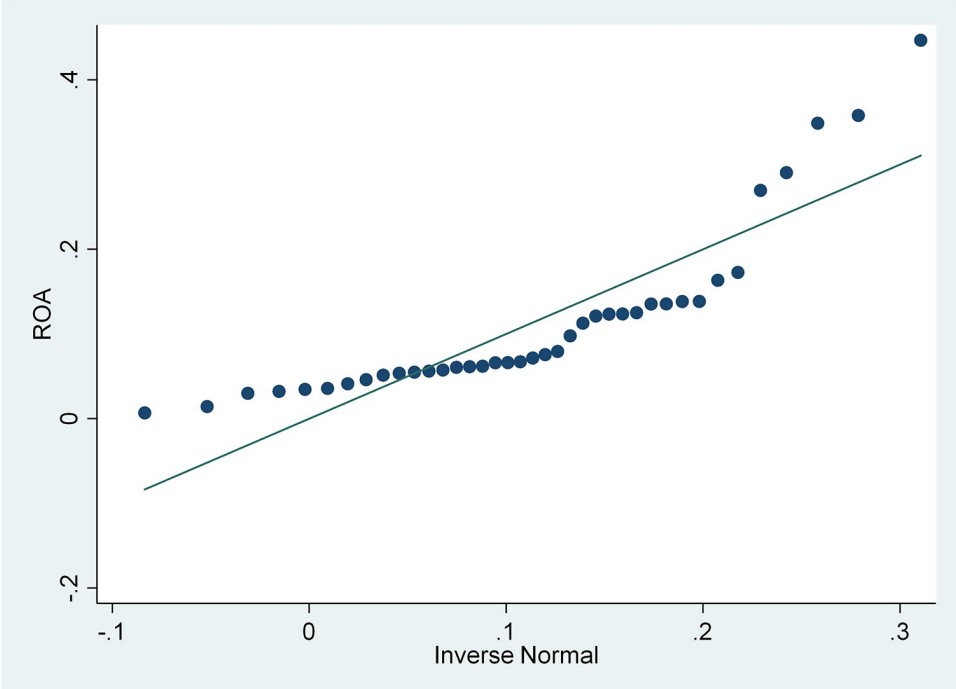

**Fig 3. Normal quantile plot (ROA).** Source: output from STATA, 2022.

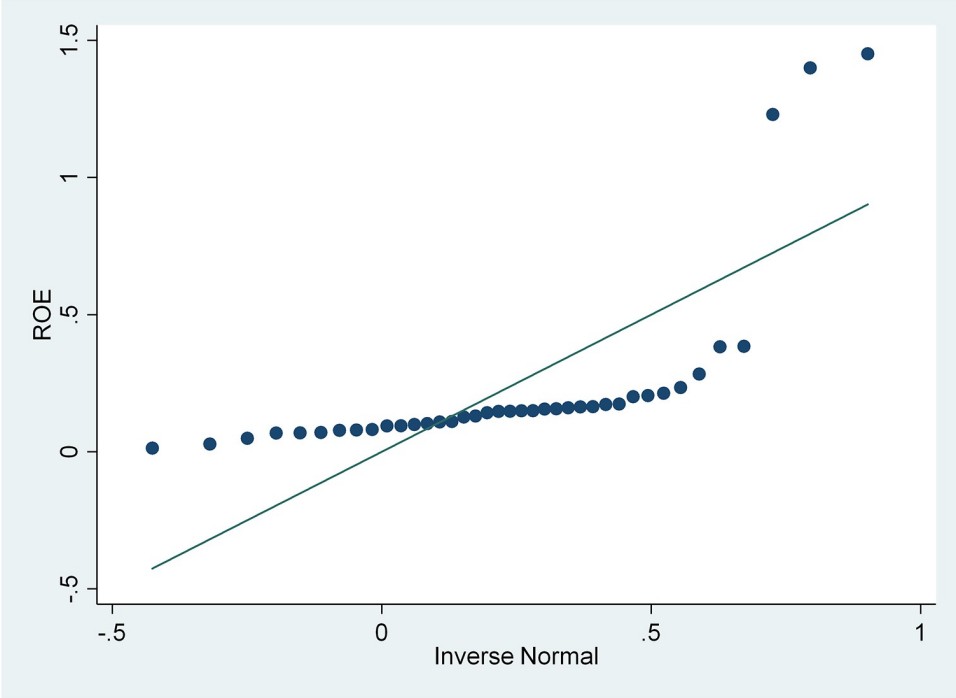

**Fig 4. Normal quantile plot (ROE).** Source: output from STATA, 2022.

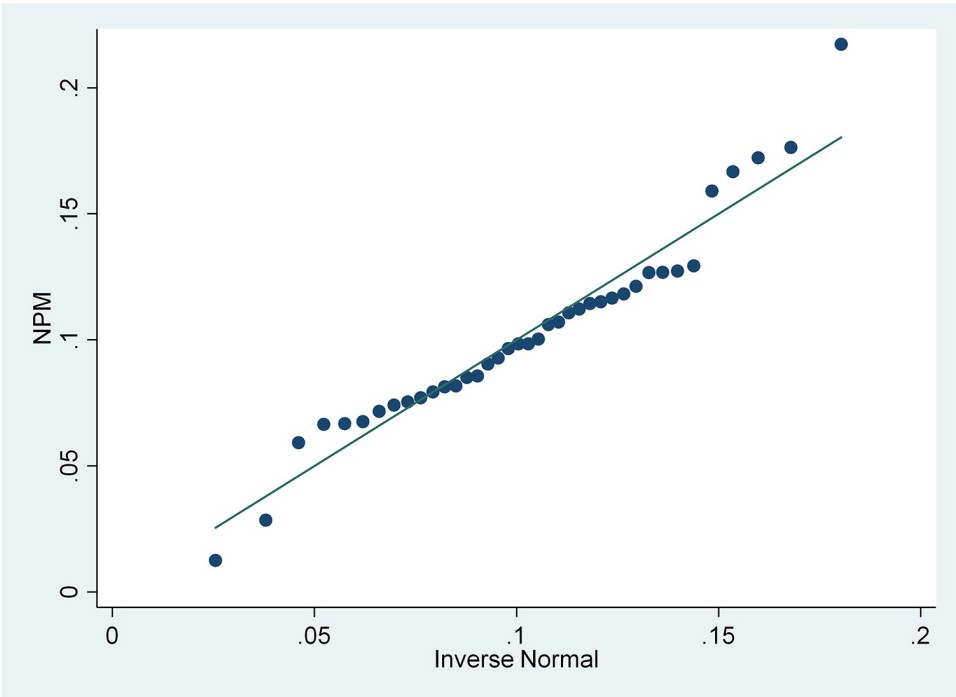

**Fig 5. Normal quantile plot (NPM).** Source: output from STATA, 2022.

**Heteroscedasticity test.**    A heteroscedasticity test was carried out to find out whether there were any dissimilarities/differences between variances or residues from the regression model that had been made [50]. In this study, the heteroscedasticity test was carried out with the white's test with the following result in Tables 5–7.

The conclusion that can be drawn from all the results of the white's test that has been tried is that the p-value is more than 0.05 so there are no symptoms of heteroscedasticity in the existing regression model.

**Autocorrelation test.**   The autocorrelation test was conducted to determine whether the results of the data collected and processed were influenced (there was a correlation) by the results of the previous year or not [50]. The autocorrelation test cannot be carried out in this study because autocorrelation test can usually only be applied to data in the form of time series, while the structure of research data is not in the form of time series.

## Hypothesis testing with coefficient of determination test, t test, F test

This test determines how well the independent variable can explain the dependent variable [52]. The t test determines how much influence each independent variable has on the

**Table 4.  Skewness and Kurtosis normality test results (transformation).**

| Variable | Obs | Pr (Skewness) | Pr (Kurtosis) | Adj chi2 (2) | Prob>chi2 |
|----------|-----|---------------|---------------|--------------|-----------|
| ln_ROA | 39 | 0.3403 | 0.1729 | 2.97 | 0.2263 |
| ln_ROE | 39 | 0.1564 | 0.0293 | 6.24 | 0.0442 |

Source: output from STATA, 2022

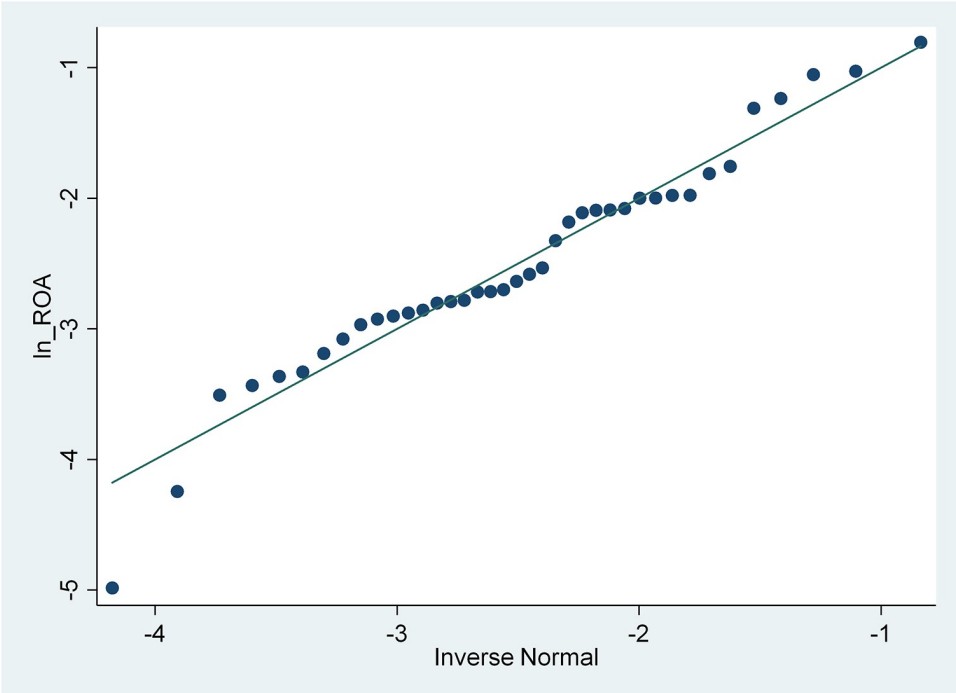

**Fig 6. Normal quantile plot (ROA after transformation).** Source: output from STATA, 2022.

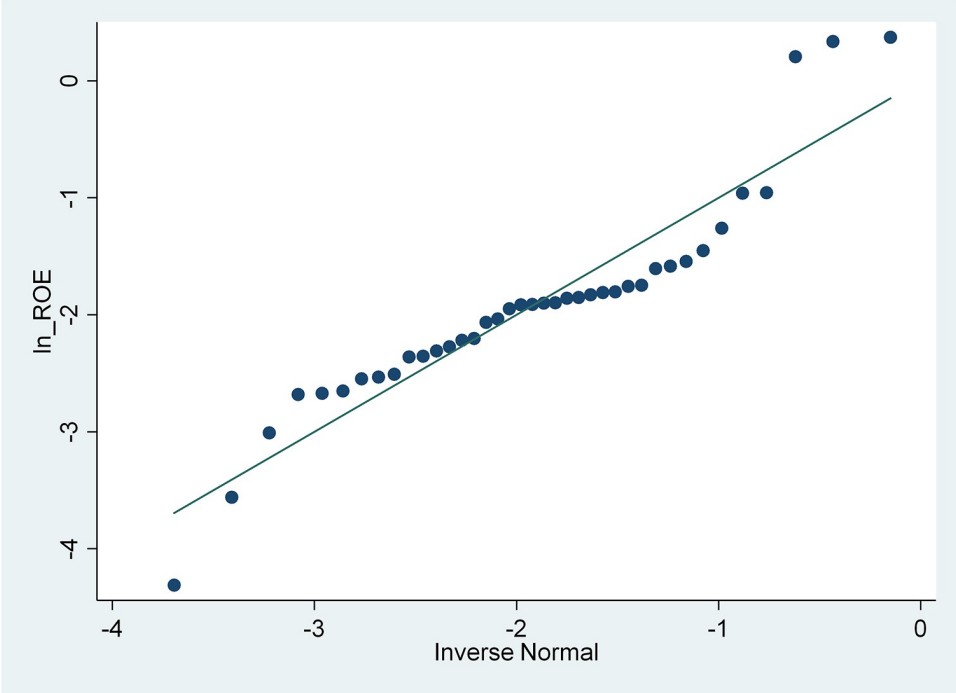

**Fig 7. Normal quantile plot (ROE after transformation).** Source: output from STATA, 2022.

**Table 5. White's test results (ROA).**

| White's test for Ho | Homoskedasticity |
|---|---|
| Against Ha | Unrestricted Heteroskedasticity |
| Chi2(2) | 3.20 |
| Prob > chi2 | 0.2024 |

Source: output from STATA, 2022

**Table 6. White's test results (ROE).**

| White's test for Ho | Homoskedasticity |
|---|---|
| Against Ha | Unrestricted Heteroskedasticity |
| Chi2(2) | 3.98 |
| Prob > chi2 | 0.1367 |

Source: output from STATA, 2022

**Table 7. White's test results (NPM).**

| White's test for Ho | Homoskedasticity |
|---|---|
| Against Ha | Unrestricted Heteroskedasticity |
| Chi2(2) | 5.18 |
| Prob > chi2 | 0.0748 |

Source: output from STATA, 2022

dependent variable, while the F test determines how much influence each independent variable has on the dependent variable concurrently [54]. Moreover, because this study employs three simple regression models, the results of the t test and F test will be identical in terms of significance.

**H1/Hypothesis 1 testing: The Effect of CSR on ROA.** In Table 8, the coefficient of determination test, the r-squared value is 0.1258, which means that the ROA variable is explained by the CSR variable of 12.58%. While the remaining 87.42% means that the ROA variable is explained by other variables outside the research regression model. In the t-test and F-test, the p-value of 0.027 is smaller than the 0.05 level of significance, which means the null hypothesis is rejected. This proves that CSR has a significant effect on ROA, so hypothesis 1 which says "CSR has a significant effect on ROA" is accepted.

**H2/Hypothesis 2 testing: The effect of CSR on ROE.** In Table 9, the coefficient of determination test, the r-squared value is 0.0494 which means that the ROE variable is explained by the CSR variable of 4.94%. While the remaining 95.06% means that the ROE variable is explained by other variables outside the research regression model. In the t-test and F-test, the p-value of 0.174 is greater than the 0.05 significance level, which means the null hypothesis is accepted. This proves that CSR does not have a significant effect on ROE, so hypothesis 2 which says "CSR has a significant effect on ROE" is rejected.

**H3/Hypothesis 3 testing: The effect of CSR on NPM.** In Table 10, the coefficient of determination test, the r-square value is 0.0021, which means that the NPM variable is explained by the CSR variable of 0.21%. While the remaining 99.79% means that the NPM variable is explained by other variables outside the research regression model. In the t-test and F-

**Table 8. Hypothesis 1 testing results.**

| Source | SS | | df | | | MS | |
|---|---|---|---|---|---|---|---|
| Model | 3.47295668 | | 1 | | | 3.47295668 | |
| Residual | 24.1272383 | | 37 | | | 0.652087522 | |
| Total | 27.600195 | | 38 | | | 0.726320921 | |
| ln_ROA | Coef. | Std. Err. | t | P>[t] | | [95% Conf. Interval] | |
| ln_CSRI | -0.5956638 | 0.2581098 | -2.31 | 0.027 | | -1.118644 | -0.0726837 |
| _cons | -3.666097 | 0.5188872 | -7.07 | 0.000 | | -4.717462 | -2.614731 |
| Number of obs | 39 | | | | | | |
| F (1,37) | 5.33 | | | | | | |
| Prob > F | 0.0267 | | | | | | |
| R-squared | 0.1258 | | | | | | |
| Adj R-squared | 0.1022 | | | | | | |
| Root MSE | 0.80752 | | | | | | |

Source: output from STATA, 2022

**Table 9. Hypothesis 2 testing results.**

| Source | SS | | df | | | MS | |
|---|---|---|---|---|---|---|---|
| Model | 1.53420595 | | 1 | | | 1.53420595 | |
| Residual | 29.5348863 | | 37 | | | 0.79824017 | |
| Total | 31.0690923 | | 38 | | | 0.817607691 | |
| ln_ROE | Coef. | Std. Err. | t | P>[t] | | [95% Conf. Interval] | |
| ln_CSRI | -0.3959071 | 0.2855738 | -1.39 | 0.174 | | -0.9745345 | 0.1827203 |
| _cons | -2.692599 | 0.574099 | -4.69 | 0.000 | | -3.855835 | -1.529364 |
| Number of obs | 39 | | | | | | |
| F (1,37) | 1.92 | | | | | | |
| Prob > F | 0.1739 | | | | | | |
| R-squared | 0.0494 | | | | | | |
| Adj R-squared | 0.0237 | | | | | | |
| Root MSE | 0.89344 | | | | | | |

Source: output from STATA, 2022

test, the p-value of 0.781 is greater than the 0.05 significance level, which means the null hypothesis is accepted. This proves that CSR does not have a significant effect on NPM, so hypothesis 3 which says "CSR has a significant effect on NPM" is rejected.

## Simple linear regression equation

**CSR and ROA.** Based on Table 11, the analysis of the simple linear regression results shown, the linear regression equation can be made:

$Y_1$(ROA) =

**CSR and ROE.** Based on Table 12, the analysis of the simple linear regression results shown, the linear regression equation can be made:

$Y_2$(ROE) =

**CSR and NPM.** Based on Table 13, the analysis of the simple linear regression results shown, the linear regression equation can be made:

$Y_3$(NPM) =

**Table 10.  Hypothesis 3 testing results.**

| Source | SS | | df | MS | |
|---|---|---|---|---|---|
| Model | 0.000125454 | | 1 | 0.000125454 | |
| Residual | 0.059135779 | | 37 | 0.001598264 | |
| Total | 0.059261232 | | 38 | 0.001559506 | |
| **NPM** | **Coef.** | **Std. Err.** | **t** | **P>[t]** | **[95% Conf. Interval]** |
| CSRI | 0.0243271 | 0.0868307 | 0.28 | 0.781 | -0.1516086 | 0.2002628 |
| _cons | 0.0990211 | 0.0153226 | 6.46 | 0.000 | 0.0679745 | 0.1300677 |
| **Number of obs** | 39 | | | | | |
| **F (1,37)** | 0.08 | | | | | |
| **Prob > F** | 0.7809 | | | | | |
| **R-squared** | 0.0021 | | | | | |
| **Adj R-squared** | -0.0249 | | | | | |
| **Root MSE** | 0.03998 | | | | | |

Source: output from STATA, 2022

**Table 11.  Simple linear regression.**

| ln_ROA | Coef. |
|---|---|
| ln_CSRI | -0.5956638 |
| cons | -3.666097 |

Source: output from STATA, 2022

**Table 12.  Simple linear regression.**

| ln_ROE | Coef. |
|---|---|
| ln_CSRI | -0.3959071 |
| cons | -2.692599 |

Source: output from STATA, 2022

**Table 13.  Simple linear regression.**

| NPM | Coef. |
|---|---|
| CSRI | 0.0243271 |
| cons | 0.0990211 |

Source: output from STATA, 2022

## 5. Discussion

The results of hypothesis 1 test proved that CSR has a significant effect on ROA. This is concluded from the analysis that the results support the theoretical view of the relationship between CSR and financial performance. However, the results of hypothesis 2 and 3 test proved that CSR has no effect on ROE and NPM. This study only used industrial manufacturing companies in the LQ45 index as research samples, so that the sample obtained was only 13 companies and the research was only carried out over a period of 3 years which makes the

number of samples as many as 39 samples. In addition, the variable used was only 1 independent variable, namely CSR, and 3 dependent variables, namely ROA, ROE, and NPM. Based on the results of research conducted on the annual report and sustainability report published online in 13 sample companies of manufacturing companies, the data for each company that makes up the 13 companies as the study sample for three consecutive years (2018–2020), as follow:

## Astra International Tbk (ASII)

The form of CSR carried out by ASII, namely (1) Cause-Related Marketing, the company is committed to making donations or contributions on a particular issue based on product sales. The company will provide financial assistance in the form of a certain percentage of sales revenue. Usually carried out within a certain period for a certain product and in the form of a certain donation. This program has two objectives, namely to obtain a certain amount of funds to donate, besides increasing product sales. This type of activity has the same goal as cause promotion, but is linked to consumer response to sales (for example, the amount of a passenger's donation is associated with the number of miles traveled by a certain company's aircraft) and (2) Socially Responsible Business Practice, this activity adopts and takes the initiative to carry out business practices and investments that support existing social problems. The nature of this activity is to do things that exceed what is required by existing laws and regulations and what is expected (discretionary) for the community such as employees, distributors, suppliers, non-profit partners, and likewise as members of the general public. While the field of activity includes health and safety, as well as emotional and psychological needs. Currently, the practice of corporate governance has shifted from dealing with customer complaints, and overcoming pressure from pressure groups to activities that are proactive in seeking solutions to existing social problems. In general, these activities are dominated by manufacturing, technology, and agricultural industry activities, where decisions are made regarding the supply chain, raw materials, operational procedures, and employee safety.

## Charoen Pokphand Indonesia Tbk (CPIN)

CPIN achieved a net profit of IDR 3.62 trillion in 2021, a decrease from the previous year's net profit of IDR 3.85 trillion. CPIN's ability to sustain this level of performance is the result of its diligence and perseverance in managing its operational activities. Throughout the reporting period, CPIN adhered to the same approach and strategy as the prior year. This decision was made considering the ongoing COVID-19 pandemic, and the implemented strategies have proven effective at maintaining the company's business performance. Social performance (people) CPIN is committed to supporting superior and competitive Human Resource (HR) management through an HR approach with superior character, specifically HR who are competent, committed, and contribute to the organization. CPIN pays close attention to the development of human resources in the community, particularly in producing superior and competitive livestock human resources. CPIN collaborates with universities with animal husbandry programs in various regions to foster an environment conducive to animal husbandry education that meets the demands of the industry. Through its CSR program, CPIN donated closed-house cages to the Teaching Farm. By the end of 2021, CPIN will have constructed and donated thirteen closed-house enclosures to ten Indonesian universities. Additionally, CPIN offers a grant program for entrepreneurial centers by donating Prima Freshmart (PFM) stores and their equipment to several universities. Through the Work-Based Academy (WBA) program, CPIN also offers internship opportunities to livestock graduates.

## Gudang Garam Tbk (GGRM)

CSR GGRM spent 48.4 billion Rp on CSR initiatives in 2020. Like many other businesses, GGRM runs a CSR initiative to better the community. Education, environmental preservation, health care, and social and cultural events are all focal points of the company's CSR initiatives. GGRM has established the Gudang Garam Foundation to aid financially disadvantaged students in completing their education. The foundation is also involved in the improvement of educational infrastructure in different regions of Indonesia. GGRM has launched several initiatives to lessen the strain it places on the natural world. Sustainable waste management, water conservation, and protecting natural habitats all fall under this category. The company has made contributions to the healthcare system in the form of free medical care for underserved communities and the donation of medical equipment to hospitals. Finally, GGRM has been active in the social and cultural activities sector, establishing the Gudang Garam Art Award to honor exceptional achievements in the arts and participating in other initiatives to preserve Indonesia's cultural heritage. As a whole, GGRM 's CSR initiatives hope to make Indonesia a better place by improving the quality of life for its citizens and the environment.

## HM Sampoerna Tbk (HMSP)

HMSP flagship CSR program, National Economic Recovery Program (NERP) through capacity building of Village Owned Enterprises in Tourism villages. This program is the development of a tourist village through training and managerial capacity building carried out in collaboration with the East Java government. NERP through tourism village economic recovery and water conservation. NERP through XZ Collaboration. NERP through build back economy better "HMSP supports NERP by encouraging innovative Micro, Small, and Medium Enterprises (MSMEs) to survive the pandemic and promoting awareness of social and environmental impacts.

## Indofood CBP Sukses Makmur Tbk (ICBP)

ICBP certainly also owns and implements a planned CSR program so that the intended contribution is truly optimal. In this case ICBP CSR has 5 pillars, each of which contains philosophy. ICBP owns and implements a planned CSR program. This matter is intended so that the contribution made is truly optimal. ICBP continues to continue the CSR which reflects the Company's mission of "Contributing to prosperity society and the environment in a sustainable manner". The foundation of the objectives used in implementing the Company's CSR program is to create a better life every day. ICBP owns and implements a planned CSR program.

## Indofood Sukses Makmur Tbk (INDF)

INDF is a leading processed food company in Indonesia. INDF is a total food solutions company with four strategic business groups: branded consumer products, bogasari, agribusiness, and distribution. INDF brands that have been attached to the hearts of Indonesian people for almost two decades such as instant noodles (indomie, supermi, and sarimi) wheat flour, cooking oil, margarine, and so on. INDF's brands have consistently been market leaders in their respective segments, and are known for their quality products at affordable prices. INDF's social responsibility actions are in line with its vision and mission, namely the vision of becoming a total food solutions company, and the mission. The CSR program is INDF's main commitment in helping the community and making an optimal contribution to society. INDF develops and implements various programs based on the five main pillars of the Long-term CSR philosophy: (1) Building human capital; (2) Maintaining social cohesion; (3)

Strengthening economic value; (4) Encouraging good governance; (5) Protecting the environment. The five main pillars of the Long-term CSR philosophy are manifested in the following programs: (1) Building human resources; (2). Maintaining social relations; (3). Protect the surrounding environment; (4). Increasing economic value; (5). Support good governance

### Indah Kiat Pulp & Paper Tbk (INKP)

CSR program towards its environment, the INKP is committed to supporting community life in and around the operational area. Referring to the Sustainable Development Goals (SDG), the INKP focuses its CSR on health, education, infrastructure, and community development. The INKP CSR program is carried out by considering the needs and priorities of the local community. With these various activities, the INKP hopes that the community will benefit so that they can improve the welfare and independence of the community, especially around the Company's business premises.

### Indocement Tunggal Prakarsa Tbk (INTP)

The CSR initiative of Indonesia's largest cement producer INTP, targets the areas of education, health, the environment, and community improvement. INTP has launched a new initiative called Kampung Batu to boost the standard of education in Indonesia's outlying regions. Teachers receive professional development opportunities, and program participants help construct and renovate educational facilities. INTP's health initiatives include giving free care to underserved communities, funding awareness campaigns, and constructing new medical facilities. To lessen its impact on the environment, the company has implemented several sustainability initiatives in the environmental sector, including cutting back on carbon emissions, boosting the use of renewable energy, and establishing water conservation and waste management initiatives. Finally, INTP has taken part in a wide range of community development activities, including the construction of infrastructure, the encouragement of small and medium-sized businesses, and the creation of new employment opportunities. CSR initiatives at INTP are driven by a desire to improve the lives of Indonesians and the planet.

### Kalbe Farma Tbk (KLBF)

KLBF's mission is to expand healthcare coverage to Indonesia's most underprivileged populations. The organization has initiated a program to supply people in underserved areas with free access to medical consultations, health education, and medication. Through expanding access to primary care, this program hopes to raise the health standards of underserved communities. KLBF is concerned about the health of the environment and has taken steps to lessen its negative effects on the planet. The business has set up programs for recycling, conserving water, and cutting down on carbon emissions. To sum up, KLBF is dedicated to implementing CSR programs that benefit the neighborhoods where it does business. Some of the company's goals are to improve access to health care, education, the status of women, and the sustainability of the environment.

### Semen Indonesia Tbk (SMGR)

CSR of SMGR such as a commitment to improve environmental quality, by participating in efforts to mitigate greenhouse gases, increasing the use of alternative energy, managing, and utilizing recycled materials, implementing innovation in all operational activities, and implementing greening programs as well as environmental conservation in a structured and planned manner.

### Sri Rejeki Isman Tbk (SRIL)

SRIL is dedicated to helping grow the neighbourhoods where it does business. Scholarships and new school buildings are just two examples of how the company is helping local kids get a good education. SRIL is dedicated to minimizing its negative effects on the environment. The business has taken steps to reduce its use of water and garbage. SRIL has implemented an environmental management system to guarantee that its business practices are in line with environmental laws and standards. SRIL has invested USD 485.94 thousand in community development programs. This investment is channeled through several programs including education, health, and agricultural programs, to the MSMEs program. All of these programs are aimed at being able to spur an increase in the quality of life of the surrounding community. As an industry that involves a large number of HR, to continue to maintain and empower the quality of SRIL employees, SRIL has employee training programs for both soft skills and hard skills through the Sritex Employee Training Institute (LPK). In 2021, Sritex LPK has provided training to 2,396 people with a total of 310,000 hours of training. SRIL also pays special attention to the Company's Occupational Health and Safety (OHS) performance. SRIL is committed to continuing to reduce the number of work accidents through several work safety programs. In 2021, the Company's OHS performance is shown to improve. The number of work accidents decreased by 10% compared to the previous year.

### Chandra Asri Petrochemical Tbk (TPIA)

CSR programs implemented by TPIA address both social and environmental concerns. The management at PTIA is aware of the importance of minimizing the harm their operations cause to the natural world. Among the company's many eco-friendly initiatives are emission and waste reduction plans, recycling initiatives, and help for community conservation initiatives. Aside from that, they provide community members with sustainability-focused education programs. As a whole, TPIA's CSR efforts show the company's dedication to making a positive difference in the world while also benefiting the company's shareholders.

### Unilever Indonesia Tbk (UNVR)

UNVR has several CSR programs, which have been carried out for people's lives. The form of CSR carried out by UNVR is by establishing the Unilever Indonesia Foundation. This foundation was formed on November 27, 2000, and was built according to Unilever's mission, namely to empower the potential that exists in Indonesian society, provide a positive added value to society, and unite strength between partners who act as a catalyst in forming a partnership. In addition, the purpose of the Unilever Indonesia Foundation was to improve the quality and quantity of people's lives. The ongoing campaign pepsodent school program to elementary schools in understanding the importance of oral health, as well as instilling understanding in children about the importance of visiting the dentist regularly. In preserving the environment, Unilever makes preservation of water sources in the Brantas River.

### The effect of CSR on ROA

In this study, it is known that CSR has a significant influence on ROA in manufacturing companies at LQ45. The findings of this study are supported by research conducted by Oyewumi et al. [43], Purwabangsa et al. [37], and Purnaningsih [44]. Manufacturing companies that adopt environmentally friendly practices can help companies reduce their operating costs. For example, adopting energy-efficient processes, reducing waste and emissions, and applying circular economy principles can help reduce a company's use of energy, water, and raw materials,

thereby lowering company costs. This can increase profits, which in turn can increase ROA. CSR initiatives can help improve the reputation and brand image of a manufacturing company. For example, adopting socially responsible employment practices, such as providing fair pay and safe working conditions and promoting diversity and inclusion, can help attract and retain the best talent. This can increase employee morale and productivity, leading to increased efficiency and higher-quality products. In addition, adopting socially responsible practices can help improve a company's reputation with customers and other stakeholders, leading to increased sales and profits and thereby increasing ROA. Implementing sustainable supply chain practices can help manufacturing companies improve efficiency, reduce costs, and manage risk. For example, sourcing raw materials from sustainable sources can help reduce supply chain disruptions and regulatory risks. Working with suppliers who practice environmentally and socially responsible practices, on the other hand, can help reduce the company's reputational risk and avoid legal liability. This can help increase a company's profits, leading to a higher ROA. Manufacturing companies that adopt CSR initiatives such as sustainable practices, socially responsible labor practices, and sustainable supply chain practices can benefit from increased efficiency, cost savings, and an enhanced reputation [20]. These benefits can result in increased revenue, higher profits, and increased ROA.

## The effect of CSR on ROE

The results of this study show that CSR has no significant effect on ROE in manufacturing companies at LQ45. The findings of this study are supported by research by Rosiliana et al. [47], which indicates that the more companies engage in CSR, the less profit they generate, resulting in a lower ROE. However, the results of this study are contradicted by Purbawangsa et al. [37] and Purnaningsih [44]. ROE measures a company's profitability relative to the amount of equity it has, which represents shareholder investment in the company. The main reason why CSR may have an insignificant impact on ROE is that it does not directly increase a company's profitability or financial performance in the short term. Conversely, CSR initiatives may require significant investment and take time to turn a profit. Implementing sustainable practices in manufacturing companies may require investing in new technologies, equipment, or processes that may increase costs in the short term. Likewise, investing in socially responsible employment practices may involve higher labor costs or expenses related to training, which may not directly contribute to profitability. Therefore, the short-term impact of CSR initiatives on company profitability may be limited. In addition, other factors such as market competition, macroeconomic conditions, and the company's financial structure can also affect ROE. For example, if a manufacturing company operates in a highly competitive market, the impact of CSR initiatives on its profitability may be limited. Likewise, if a company has a high level of debt or a weak financial structure, the impact of CSR initiatives on its financial performance may be limited. However, although the direct impact of CSR on ROE may be limited, it can indirectly contribute to long-term profitability and shareholder value. For example, adopting sustainable practices can help companies reduce costs and increase efficiency, leading to increased profitability in the long run. In addition, adopting socially responsible practices can improve a company's reputation and brand image, which can contribute to increased sales and customer loyalty over time. These benefits can ultimately contribute to increased shareholder value and ROE.

## The effect of CSR on NPM

The research results show that CSR has no significant effect on NPM in manufacturing companies at LQ45. This research is supported by Rosiliana et al. [47], who states that engaging in

CSR activities can improve a company's reputation and increase public interest in purchasing its products, but it does not guarantee an increase in NPM. CSR initiatives do not directly increase a company's NPM in the short term. NPM measures a company's profitability relative to its revenue. While CSR initiatives may improve a company's reputation and enhance its brand image, which may attract customers and increase sales, the impact on NPM may not be significant. Implementing CSR initiatives such as sustainable practices or socially responsible labor practices may require investments and expenses that may increase costs and reduce net profit margins in the short term. Investing in renewable energy sources, reducing waste, or improving working conditions may require upfront investments that may increase costs in the short term. Adopting fair labor practices may require higher wages or benefits, which may reduce NPM in the short term. However, these investments and expenses may ultimately contribute to increased efficiency, productivity, and sales, leading to improved profitability in the long term. Other factors such as market competition, macroeconomic conditions, and the company's financial structure can also influence NPM. A manufacturing company operates in a highly competitive market, so the impact of CSR initiatives on its NPM may be limited. Similarly, if the company has high levels of debt or a weak financial structure, the impact of CSR initiatives on its financial performance may be limited. While the direct impact of CSR on NPM may be limited, it can indirectly contribute to long-term profitability and sustainability. Adopting CSR initiatives can help a manufacturing company enhance its reputation, attract customers and talent, and reduce risks, which can contribute to increased sales, productivity, and innovation over time. These benefits can ultimately contribute to improved profitability and long-term sustainability.

## Policy recommendations

In Indonesia, CSR is important not only for ethical and social reasons but also for financial performance, especially for manufacturing companies. Indonesian regulations mandate that companies engage in CSR initiatives as part of their business operations. Failure to comply with these regulations can result in legal and financial penalties. Prioritizing CSR can help companies meet these requirements and avoid legal and financial risks. Increase your operational efficiency: Adopting CSR initiatives, such as sustainable practices or responsible supply chain management, can improve a company's operational efficiency, reduce costs, and increase productivity. This can improve profitability and financial performance, which are critical in Indonesia's highly competitive manufacturing sector. Manufacturing companies in Indonesia that prioritize CSR can benefit by meeting regulatory requirements, enhancing their reputation and brand image, increasing operational efficiency, accessing new markets, and meeting stakeholder expectations. These benefits can ultimately contribute to improved financial performance and long-term sustainability. Therefore, manufacturing companies in Indonesia need to prioritize CSR as a strategic policy recommendation for long-term financial success.

## 6. Conclusion

Legislation in Indonesia already regulates CSR that is how companies manage social and environmental responsibility. The implementation of CSR in manufacturing companies in Indonesia is still not running properly because there are still many companies that have not implemented CSR. CSR publication causes companies to incur significant additional costs. Expenses will undoubtedly impact the company's profit. On the other hand, it will also enhance the company's reputation in the eyes of the public. so that the community responds positively to the company's products, which has an effect on the company's bottom line. If profit increases, then ROA will increase as well. CSR can increase public confidence in the

company's products, thereby enhancing the company's reputation in the eyes of the public. Therefore, consumers will desire the company's products. The greater the market success of the company's products, the greater the profit that the company can generate. Suggestion for companies, of course, it is better to immediately disclose CSR in the annual report or make their own sustainability report specifically so that company stakeholders can see for themselves how the company carries out CSR activities that are useful for the benefit of society and the environment. It is hoped that the company can publish CSR activities carried out, because CSR can provide several benefits for companies such as providing a good product image so that it can increase return on assets, return on equity, and net profit margin and enhance the company's reputation. This also proves that the company is not only concerned with the profits it receives but also with the community and the surrounding environment. For further research, researchers should use more samples by expanding the research population and extending the period of the study so that more samples are tested and studied than before. The variables used can also be changed to other dependent variables or directly choose the independent variable more than 1 and only one dependent variable so that multiple regression analysis can be carried out in future research.

## 7. Limitation & implication

This study only involved a few manufacturing companies and only companies listed on LQ45. The short period of this study is also its main research limitation, making it difficult to generalize to the year or make precise estimates. This research has implications, especially for manufacturing companies formulating strategies through CSR. The right CSR program will improve the company's financial performance. This research provides an understanding and description of the impact of good CSR on the company, so the company must formulate a CSR program that can benefit society in Indonesia.

## Author Contributions

**Conceptualization:** Gatot Soepriyanto, Mochammad Fahlevi, Sandra Grabowska, Mohammed Aljuaid.

**Data curation:** Meiryani, Jessica, Mochammad Fahlevi, Mohammed Aljuaid.

**Formal analysis:** Meiryani, Shi Ming Huang, Gatot Soepriyanto, Mochammad Fahlevi, Sandra Grabowska.

**Funding acquisition:** Sandra Grabowska.

**Investigation:** Meiryani, Shi Ming Huang, Jessica, Mohammed Aljuaid.

**Methodology:** Shi Ming Huang, Sandra Grabowska, Mohammed Aljuaid.

**Project administration:** Meiryani.

**Resources:** Meiryani, Shi Ming Huang.

**Software:** Meiryani, Shi Ming Huang, Gatot Soepriyanto, Jessica.

**Supervision:** Meiryani, Shi Ming Huang, Gatot Soepriyanto.

**Validation:** Meiryani, Gatot Soepriyanto, Sandra Grabowska.

**Visualization:** Shi Ming Huang, Jessica.

**Writing – original draft:** Meiryani, Shi Ming Huang, Mochammad Fahlevi.

**Writing – review & editing:** Meiryani, Gatot Soepriyanto, Mochammad Fahlevi, Sandra Grabowska, Mohammed Aljuaid.

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
