## [Decision Letter · Decision Letter 0]

8 Feb 2023

PONE-D-22-31646

The Effect of Corporate Social Responsibility on Financial Performance in LQ45 Manufacturing Companies

PLOS ONE

Dear Dr. Meiryani,

Thank you for submitting your manuscript to PLOS ONE. After careful consideration, we feel that it has merit but does not fully meet PLOS ONE’s publication criteria as it currently stands. Therefore, we invite you to submit a revised version of the manuscript that addresses the points raised during the review process.

We look forward to receiving your revised manuscript.

Kind regards,

Simon Grima, PhD

Academic Editor

PLOS ONE

Journal Requirements:

2. PLOS ONE does not copy edit accepted manuscripts (https://journals.plos.org/plosone/s/criteria-for-publication#loc-5). To that effect, please ensure that your submission is free of typos and grammatical errors.

5. We note you have included a table to which you do not refer in the text of your manuscript. Please ensure that you refer to Table 4.1 to 4.9 in your text; if accepted, production will need this reference to link the reader to the Table.

Additional Editor Comments:

Please review your paper according to the comments of the reviewers and send a covering letter explaining how the review was carried out

Reviewer 1

• The title must be complete by adding ‘in Indonesia’ to make it complete

• The abstract must have a conclusion and recommendations of the study

• The last paragraph of your introduction section must end up by giving clues about the organisation of the paper. Revise.

• The literature section must capture empirical literature on CSR Strategies, CSR Disclosures and CSR initiatives and activities

• How the data were extracted from the Annual Reports must be captured for replication purposes. That is the procedure. The Data Description and the measurement of the variables must be reported.

• The result section must report the data for each company that makes up the 13 companies as the study sample. This will be insightful for readership and analysis. Putting all the 13 companies as an entity will not reveal the undertones of the individual company’s

• These references must be deleted from the reference list as they did not appear in the text. Reference numbers 5, 9, 14, 15 and 16.

• Incomplete reference sources, either with page numbers for the journals or the absence of the location of the publisher for books. Reference numbers 4, 6, 7, 10, 11, 17, 18, 24 and 25

• The author(s) must do well to improve the quality of English in the paper

Reviewer 2

The Effect of Voluntary Disclosure on Financial Performance: Empirical Study on Manufacturing Companies in Indonesia

Abstract

1. The first line disappointed me; rather, it should be like this…. "The obedience of the manufacturing sector industry to manage natural resources from the environment still needs to be higher.

2. There is a need to improve the style of writing.

3. Without highlighting the purpose of the study, the authors have straight forwarded moved to the research methodology. I am not happy with the English part of this manuscript in the abstract section.

Introduction

1. Background of the study is missing. I cannot see any outline or an overview of the topic

2. What is the source of this? According to the Ministry of Environment and Forestry (KLHK), the compliance of the manufacturing sector industry in managing natural resources from the environment is still low. Until 2019, the number of companies that registered themselves to be assessed for compliance was quite low, only reaching 597 companies or 29.15 per cent."

3. I do not understand this” In this research does not expand and is more focused on the research objectives that have been described, the research limitations are…."

4. Conducted on manufacturing industry companies: can you please define why you have selected manufacturing industry?

5. The author said “manufacturing industry companies” is this correct way of writing?

Literature review

1. It could be more satisfactory. Please focus on the research gap .. Strengthen the argument of your study by referring to a larger body of scientific literature, and clearly outline what is missing in the literature and what gap you wish to fill in. Why is the focus only on old theories? Kindly add relevant latest literature since CSR is the talk of the town. It is a part of financial disclosure.

2. What are the selection criteria for this 91 indicator? Please clarify.

Research methodology

1. Please elaborate on how and why purposive sampling.

2. What was the population size here?

3. In the Data Collection Method, who were the respondents

Conclusion

1. I can’t see much difference between discussion and conclusion part. Actually the author has written a small portion of the analysis and discussion part in the conclusion.

Kindly rewrite the entire conclusion section. There is a need for the division of discussion and conclusion into two separate sections, with the conclusion section simply needing to give the most critical information in points so that readers can quickly find what they want from the publication. The discussion section might need to add more subheadings, such as a discussion of the findings, main contributions, policy recommendations, limitations, and future work of this study. The manuscript's logic would substantially increase if the authors were willing to modify it.

References

1. Many mistakes in references like volume nos and page nos etc. are missing, for example: 23-25, 3, 5, 4, 7, 8. there is a need to go through all the references carefully. Please check all the references.

2. In text style of citation is also not properly followed. Mistakes at many places and need to add more citations

Reviewers' comments:

Reviewer's Responses to Questions

**Comments to the Author**

1. Is the manuscript technically sound, and do the data support the conclusions?

Reviewer #1: Yes

Reviewer #2: Partly

2. Has the statistical analysis been performed appropriately and rigorously? 

Reviewer #1: Yes

Reviewer #2: Yes

3. Have the authors made all data underlying the findings in their manuscript fully available?

Reviewer #1: Yes

Reviewer #2: No

4. Is the manuscript presented in an intelligible fashion and written in standard English?

Reviewer #1: Yes

Reviewer #2: No

5. Review Comments to the Author

Reviewer #1: •The title must be complete by adding ‘in Indonesia’ to make it complete

•The abstract must have a conclusion and recommendations of the study

•The last paragraph of your introduction section must end up by giving clues about the organisation of the paper. Revise.

•The literature section must capture empirical literature on CSR Strategies, CSR Disclosures and CSR initiatives and activities

•How the data were extracted from the Annual Reports must be captured for replication purposes. That is the procedure. The Data Description and the measurement of the variables must be reported.

•The result section must report the data for each company that makes up the 13 companies as the study sample. This will be insightful for readership and analysis. Putting all the 13 companies as an entity will not reveal the undertones of the individual company’s

•These references must be deleted from the reference list as they did not appear in the text. Reference numbers 5, 9, 14, 15 and 16.

•Incomplete reference sources, either with page numbers for the journals or the absence of the location of the publisher for books. Reference numbers 4, 6, 7, 10, 11, 17, 18, 24 and 25

•The author(s) must do well to improve the quality of English in the paper

Reviewer #2: The Effect of Voluntary Disclosure on Financial Performance: Empirical Study on Manufacturing Companies in Indonesia

Abstract

1.The first line disappointed me; rather, it should be like this…. "The obedience of the manufacturing sector industry to manage natural resources from the environment still needs to be higher.

2.There is a need to improve the style of writing.

3.Without highlighting the purpose of the study, the authors have straight forwarded moved to the research methodology. I am not happy with the English part of this manuscript in the abstract section.

Introduction

1.Background of the study is missing. I cannot see any outline or an overview of the topic

2.What is the source of this? According to the Ministry of Environment and Forestry (KLHK), the compliance of the manufacturing sector industry in managing natural resources from the environment is still low. Until 2019, the number of companies that registered themselves to be assessed for compliance was quite low, only reaching 597 companies or 29.15 per cent."

3.I do not understand this” In this research does not expand and is more focused on the research objectives that have been described, the research limitations are…."

4.Conducted on manufacturing industry companies: can you please define why you have selected manufacturing industry?

5.The author said “manufacturing industry companies” is this correct way of writing?

Literature review

1.It could be more satisfactory. Please focus on the research gap .. Strengthen the argument of your study by referring to a larger body of scientific literature, and clearly outline what is missing in the literature and what gap you wish to fill in. Why is the focus only on old theories? Kindly add relevant latest literature since CSR is the talk of the town. It is a part of financial disclosure.

2.What are the selection criteria for this 91 indicator? Please clarify.

Research methodology

1.Please elaborate on how and why purposive sampling.

2.What was the population size here?

3.In the Data Collection Method, who were the respondents

Conclusion

1.I can’t see much difference between discussion and conclusion part. Actually the author has written a small portion of the analysis and discussion part in the conclusion.

Kindly rewrite the entire conclusion section. There is a need for the division of discussion and conclusion into two separate sections, with the conclusion section simply needing to give the most critical information in points so that readers can quickly find what they want from the publication. The discussion section might need to add more subheadings, such as a discussion of the findings, main contributions, policy recommendations, limitations, and future work of this study. The manuscript's logic would substantially increase if the authors were willing to modify it.

References

1.Many mistakes in references like volume nos and page nos etc. are missing, for example: 23-25, 3, 5, 4, 7, 8. there is a need to go through all the references carefully. Please check all the references.

2.In text style of citation is also not properly followed. Mistakes at many places and need to add more citations

The manuscript will be accepted only after major revisions

6. PLOS authors have the option to publish the peer review history of their article (what does this mean?). If published, this will include your full peer review and any attached files.

Reviewer #1: **Yes: **Charles Adusei

Reviewer #2: **Yes: **KIRAN SOOD

---

## [Author Response · Author response to Decision Letter 0]

25 Apr 2023

Response to Reviewers’

Many thanks to reviewer 1 for valuable comment

1) Reviewer 1 : The title must be complete by adding ‘in Indonesia’ to make it complete

 Authors : we have done revision please see page 1

2) Reviewer 1 : The abstract must have a conclusion and recommendations of the study

 Authors : we have done revision please see page 1

3) Reviewer 1 : The last paragraph of your introduction section must end up by giving clues about the organisation of the paper. Revise.

 Authors : we have done revision please see page 3

4) Reviewer 1 : The literature section must capture empirical literature on CSR Strategies, CSR Disclosures and CSR initiatives and activities

 Authors : we have done revision please see page 3-5

5) Reviewer 1 : How the data were extracted from the Annual Reports must be captured for replication purposes. That is the procedure. The Data Description and the measurement of the variables must be reported.

 Authors : we have done revision please see page 9-12

6) Reviewer 1 : The result section must report the data for each company that makes up the 13 companies as the study sample. This will be insightful for readership and analysis. Putting all the 13 companies as an entity will not reveal the undertones of the individual company’s

 Authors : we have done revision please see page 18-

7) Reviewer 1 : • These references must be deleted from the reference list as they did not appear in the text. Reference numbers 5, 9, 14, 15 and 16.

 Authors : we have done revision please see page 

8) Reviewer 1 : Incomplete reference sources, either with page numbers for the journals or the absence of the location of the publisher for books. Reference numbers 4, 6, 7, 10, 11, 17, 18, 24 and 25

 Authors : we have done revision please see page 26-27

9) Reviewer 1 : The author(s) must do well to improve the quality of English in the paper

 Authors : we have done revision please see page 1-28

Responses to Reviewer

Many thanks to reviewer 2 for valuable comment

Abstract

1. Reviewer : The first line disappointed me; rather, it should be like this…. "The obedience of the manufacturing sector industry to manage natural resources from the environment still needs to be higher.

Authors : we have done revision please see page 1

2. Reviewer : There is a need to improve the style of writing.

Authors : we have done revision please see page 1

3. Reviewer : Without highlighting the purpose of the study, the authors have straight forwarded moved to the research methodology. I am not happy with the English part of this manuscript in the abstract section.

Authors : we have done revision please see page 1

Introduction

1. Reviewer : Background of the study is missing. I cannot see any outline or an overview of the topic

Authors : we have done revision please see page 1

2. Reviewer : What is the source of this? According to the Ministry of Environment and Forestry (KLHK), the compliance of the manufacturing sector industry in managing natural resources from the environment is still low. Until 2019, the number of companies that registered themselves to be assessed for compliance was quite low, only reaching 597 companies or 29.15 per cent."

Authors : we have done revision please see page 2

3. Reviewer : I do not understand this” In this research does not expand and is more focused on the research objectives that have been described, the research limitations are…."

Authors : we have done revision please see page 22-23

4. Conducted on manufacturing industry companies: can you please define why you have selected manufacturing industry?

Authors : we have done revision please see page 3

5. The author said “manufacturing industry companies” is this correct way of writing?

Authors : we have done revision please see page 1-25

Literature review

1. Reviewer : It could be more satisfactory. Please focus on the research gap .. Strengthen the argument of your study by referring to a larger body of scientific literature, and clearly outline what is missing in the literature and what gap you wish to fill in. Why is the focus only on old theories? Kindly add relevant latest literature since CSR is the talk of the town. It is a part of financial disclosure.

Authors : we have done revision please see page 1-3

2. Reviewer : What are the selection criteria for this 91 indicator? Please clarify.

Authors : The 91 indicators used in this study is based on GRI G4 guidelines. we have done revision please see page 5

Research methodology

1. Reviewer : Please elaborate on how and why purposive sampling.

Authors : The reason for using this purposive sampling technique is because it is in accordance with this research, namely using quantitative research, and refer to previous research and studies that do not generalize (Uma Sekaran, 2016). we have done revision please see page 10

2. Reviewer : What was the population size here? 

Authors : namely 45 companies, we have done revision please see page 10

3. Reviewer : In the Data Collection Method, who were the respondents

Authors : in this study we use secondary data, namely data collected by www.idx.co.id, so we don't use quesioner fill by respondent, you can see in page 10-12

Conclusion

1. Reviewer : I can’t see much difference between discussion and conclusion part. Actually the author has written a small portion of the analysis and discussion part in the conclusion.

Authors : we have done revision please see page 17-23

Kindly rewrite the entire conclusion section. There is a need for the division of discussion and conclusion into two separate sections, with the conclusion section simply needing to give the most critical information in points so that readers can quickly find what they want from the publication. The discussion section might need to add more subheadings, such as a discussion of the findings, main contributions, policy recommendations, limitations, and future work of this study. The manuscript's logic would substantially increase if the authors were willing to modify it.

Authors : we have done revision please see page 17-23

References

1. Reviewer : Many mistakes in references like volume nos and page nos etc. are missing, for example: 23-25, 3, 5, 4, 7, 8. there is a need to go through all the references carefully. Please check all the references.

Authors : we have done revision please see page 23-27

2. Reviewer : In text style of citation is also not properly followed. Mistakes at many places and need to add more citations.

Authors : we have done revision please see page 1-27

Best Regards,

DR. MEIRYANI

---

## [Editor Report · Decision Letter 1]

2 May 2023

The Effect of Voluntary Disclosure on Financial Performance: Empirical Study on Manufacturing Industry in Indonesia

PONE-D-22-31646R1

Dear Dr. Meiryani,

We’re pleased to inform you that your manuscript has been judged scientifically suitable for publication and will be formally accepted for publication once it meets all outstanding technical requirements.

Kind regards,

Simon Grima, PhD

Academic Editor

PLOS ONE

Additional Editor Comments (optional):

Reviewers' comments:

<quillbot-extension-portal></quillbot-extension-portal>

---

## [Editor Report · Acceptance letter]

25 May 2023

PONE-D-22-31646R1 

The Effect of Voluntary Disclosure on Financial Performance: Empirical Study on Manufacturing Industry in Indonesia 

Dear Dr. -:

I'm pleased to inform you that your manuscript has been deemed suitable for publication in PLOS ONE. Congratulations! Your manuscript is now with our production department. 

Kind regards, 

on behalf of

Professor Simon Grima 

Academic Editor

PLOS ONE